# CLUSTERING AND ORDERING VARIABLE-SIZED SETS: THE CATALOG PROBLEM

## ABSTRACT

Prediction of a **varying number** of **ordered clusters** from sets of **any cardinality** is a challenging task for neural networks, combining elements of set representation, clustering and learning to order. This task arises in many diverse areas, ranging from medical triage, through multi-channel signal analysis for petroleum exploration to product catalog structure prediction. This paper focuses on the latter, which exemplifies a number of challenges inherent to adaptive ordered clustering, referred to further as the eponymous *Catalog Problem*. These include learning variable cluster constraints, exhibiting relational reasoning and managing combinatorial complexity. Despite progress in both neural clustering and set-to-sequence methods, no joint, fully differentiable model exists to-date. We develop such a modular architecture, referred to further as Neural Ordered Clusters (NOC), enhance it with a specific mechanism for learning cluster-level cardinality constraints, and provide a robust comparison of its performance in relation to alternative models. We test our method on three datasets, including synthetic catalog structures and PROCAT, a dataset of real-world catalogs consisting of over 1.5 M products, achieving state-of-the-art results on a new, more challenging formulation of the underlying problem, which has not been addressed before. Additionally, we examine the network's ability to learn higher-order interactions and investigate its capacity to learn both compositional and structural rulesets.

## 1 INTRODUCTION

The ability to group members of a set and order these groups is key to many important real-world decision-making processes. It finds applications ranging from supply chain management (Wenzel et al., 2019) to prioritization in medical triage (Miles et al., 2020). Other application domains include petroleum exploration (Rabiller et al., 2010), business process analytics (Le et al., 2014), and also product catalog structuring (Jurewicz & Derczynski, 2022), where the goal is to take a set of products and work out how to group them together and order these groups to form a coherent product catalog. We term this problem of simultaneously grouping and ordering a set of items the Catalog Problem.

This paper defines the Catalog Problem and presents an investigation into neural network approaches to it. To this end we introduce a fully-differentiable, deep learning (DL) model architecture that addresses the Catalog Problem. In it, sets of items are clustered into groups, and an ordering between groups is established. All of this is achieved in a *supervised* manner. While clustering methods are often unsupervised (Aljalbout et al., 2018; Ronen et al., 2022), the meaningful ordering of clusters often requires more knowledge than is available from the instance representation alone.

Similarly, learning to order is often framed as a supervised learning task (Vinyals et al., 2015; Yin et al., 2020; Shi, 2022). Referred to further as *set-to-sequence* (S2S), this area and its corresponding methods inspire the cluster-ordering aspect of our proposed Neural Ordered Clusters (NOC) model. Both neural clustering and set-to-sequence models have limitations. Element-wise neural clustering methods require $O(n)$ passes over the input set of cardinality $n$.[1] Cluster-wise and attention-based models are more computationally efficient, but exhibit a limited ability to learn cluster cardinality constraints (Pakman et al., 2020), integral to both the prototypical Catalog Problem and its practical

---

[1] $O(n)$ can be prohibitive with large input sets ($n >= 1000$), which is often the case in many interesting set-input problems such as 3D point cloud tasks (Qi et al., 2017; Ge et al., 2018; Zhao et al., 2021).

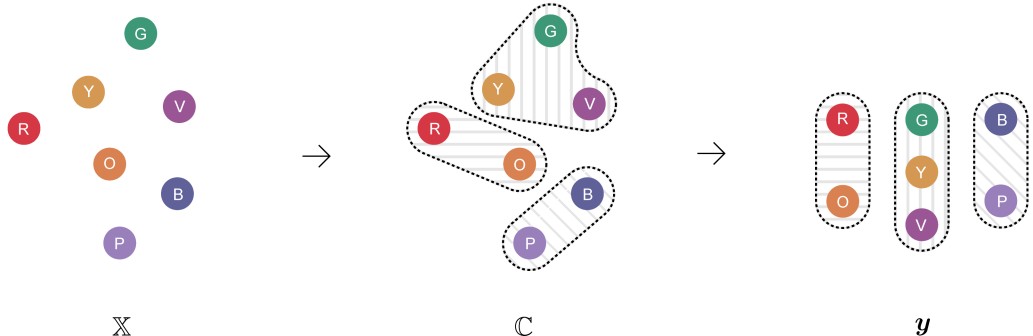

Figure 1: The Catalog Problem. From left to right: a set of input elements ($\mathbb{X}$); a clustering of those elements ($\mathbb{C}$); and a target ordering over those clustered elements ($\boldsymbol{y}$), left to right. The model has to perform all these tasks using information about inter-element relations and intra-cluster relations in order to characterise a cluster, and inter-cluster relations to generate the final, ordered clustering.

instantiations. Set-to-sequence methods, on the other hand, are effective at learning constraints (Zhu et al., 2021) and generalizing to unseen distributions (Wen, 2022). However, they are limited by their inability to predict an *adaptive* number $k$ of clusters without major adjustments (Fernández-González, 2022), two of which are proposed in Section 5.1. Nonetheless, these S2S variants suffer from noisy in-cluster order and cascading first-choice costs (Gan et al., 2020; Vial et al., 2022).

To address these challenges, we implement a unified clustering and cluster ordering method. NOC is capable of predicting ordered, partitional cluster assignments for elements of sets of varying cardinality. It infers a flexible, input-dependent number of diverse clusters, maintains $O(k)$ complexity and utilizes a jointly learned representation of set elements to find the target cluster order. Unlike existing neural clustering methods, it exhibits the ability to learn cluster cardinality constraints through supervision. To our knowledge, no other neural-based method exists to address such challenges in an end-to-end, jointly trainable way, instead performing clustering and ordering as two separate tasks, sometimes with the separate addition of a representation learning step (Aljalbout et al., 2018). All code, hyper-parameters and datasets required for reproducing our results are made available and detailed via the appendix. To summarize, our contributions are as follows:

- Firstly, we introduce the Catalog Problem, a novel joint clustering and cluster ordering problem over sets of elements, which is a challenging variant of the set-to-sequence domain with multiple aspects that are not handled by existing neural methods. We exemplify and tackle this problem on three datasets, including a real-world dataset of over 1.5 M products grouped and ordered into product catalogs by human experts.

- Secondly, we propose a novel, fully differentiable, joint neural clustering and cluster ordering model, Neural Ordered Clusters (NOC), capable of predicting an adaptive, input-dependent number of ordered, partitional clusters from sets of varying cardinality.

- Thirdly, we provide a robust comparison of existing and proposed neural methods on the Catalog Problem using synthetic & real-world datasets, providing insights into the models' capacity to learn higher-order relational rules of cluster composition and ordered structure.

## 2 THE CATALOG PROBLEM

Many problems require predicting an adaptive, input-dependent number of partitional clusters from sets of varying cardinality and consequently ordering these clusters according to a target preference. We refer to this as the Catalog Problem. In the Catalog Problem, the input is an unordered set of unique elements. The output is a clustering of these elements, with suitable cluster cardinalities, and an ordering over the clusters (Figure 1). The input may be of any cardinality. Candidate approaches to the problem have to determine how many clusters to create, choose which items to assign to which clusters and also order the clusters. This is a general problem that, as is shown by experiments later in this paper, is non-trivial.

Although the Catalog Problem is so named because it models the task of creating a catalog of items, e.g. products, no specific application is prescribed; the problem only defines input and output types and a relation between these two. The difficulty lies in learning the relationships between both input elements and groups thereof. This difficulty can be compounded by the uniqueness of input elements, making learning representations difficult, due to the scarcity of distributional information.

## 2.1 RELATED WORK

There have been many machine learning (ML) approaches to clustering with some notion of order, albeit often aimed at preventing the impact of this order on the clusters (Fisher et al., 1992). In the more common, *unsupervised* setting these range from hierarchical clustering (Johnson, 1967; Chu, 1974), through ordinal clustering (Janowitz, 1978) and incremental conceptual clustering (Fisher, 1987) to Markov clustering (Van Dongen, 2000) and other, more recent methods (Ankerst et al., 1999; Turowski et al., 2020). Certain unsupervised clustering methods without the ordering element, like affinity propagation (Frey & Dueck, 2007; Vlasblom & Wodak, 2009), are also capable of outputting an adaptive, input-dependent number of partitional clusters.

Closer to the supervised setting of interest, there have been attempts to leverage instance labels to augment k-means (Ergun et al., 2022), improve the interpretability thereof (Peng et al., 2022) and to cluster labelled data to facilitate permutation learning (Lee & Kim, 2020). Similarly, contrastive clustering utilizes soft labels to maximize the similarities of positive pairs while minimizing those of negative ones (Li et al., 2021), in an approach reminiscent of the pairwise order prediction modules that resulted in increased performance on strictly set-to-sequence tasks (Yin et al., 2020). However, these supervised clustering methods do not yield an ordering of clusters.

## 3 BACKGROUND

We identify three classes of neural approaches to solving aspects of the Catalog Problem: set representation; neural clustering; and ordering through pointer attention. Firstly, learning permutation invariant **set representations** that can encode higher-order interactions is vital, due to the complex relational factors among set elements that determine the target output. Deep learning advances in set representation focus primarily on being able to effectively learn such relations, starting with *Deep Sets* (Zaheer et al., 2017), through the *Set Transformer* (Lee et al., 2019) to modifications thereof (Girgis et al., 2021; Jurewicz & Derczynski, 2022). These methods can be used for both encoding elements and representing clusters. In the Set Transformer, given an unordered set ($\mathbb{X}$), we obtain the representations of set elements ($\boldsymbol{E}_\pi$) and subsequently the entire input set ($\boldsymbol{s}$) via:

$$\boldsymbol{E}_\pi = \mathrm{MAB}(\mathbb{X}, \mathbb{X}) = \mathrm{LN}(\boldsymbol{H} + \phi(\boldsymbol{H})), \text{ where } \boldsymbol{H} = \mathrm{LN}(\boldsymbol{X} + \mathrm{MHA}(\boldsymbol{X}_q, \boldsymbol{X}_k, \boldsymbol{X}_v)), \quad (1)$$

$$\boldsymbol{s} = \mathrm{PMA}(\boldsymbol{E}_\pi) = \mathrm{MAB}(\boldsymbol{r}, \boldsymbol{E}_\pi). \quad (2)$$

Here, multihead, intra-set attention (denoted MHA) is performed by casting the input set $\mathbb{X}$ to query, key, and value matrices $\boldsymbol{X}_q, \boldsymbol{X}_k, \boldsymbol{X}_v$ according to an arbitrary permutation $\pi$, and adding a residual connection as defined by Vaswani et al. (2017), without positional encoding. This operation is incorporated into a multihead self-attention block (MAB) by the inclusion of a row-wise feed-forward neural network (NN) $\phi$, with layer normalization (LN) after each block (Ba et al., 2016), resulting in a permutation equivariant[2] matrix of per-element representations ($\boldsymbol{E}_\pi$). These are then aggregated into a permutation invariant representation of the entire set ($\boldsymbol{s}$) by performing pooling by multihead attention (PMA) between the per-element representations and a learned seed vector $\boldsymbol{r}$. These operations are used extensively in our method for encoding both the initial input set and the predicted clusters.

Secondly, supervised **neural clustering** obtains per-element cluster assignments ($\hat{c}_i$) through a number of modular functions parameterized by NNs. These networks leverage set representation methods to encode the set of currently available, unassigned elements ($\boldsymbol{U}_j$), each previously completed cluster ($\boldsymbol{g}_j$) and consequently all clustered elements jointly ($\boldsymbol{G}_j$), at each step $j$. This is paired

---

[2]For a formal proof, see Section 3.1 and supplementary material of Lee et al. (2019).

with an algorithm for selecting the next $j$-th cluster (if clusterwise) or element (if pointwise) to be considered until nothing remains to be assigned.

In the $O(k)$ clusterwise formulation each cluster assignment is the output of another NN ($\rho$), in the form of a binary choice ($\hat{c}_i$) per encoded element ($\boldsymbol{x}_i$), conditioned on these representations and trained in a teacher-forced manner, with loss calculated only for the elements belonging to the current cluster. In the attention-based, clusterwise framework of the *Attentive Clustering Process* (Pakman et al., 2020) a random anchor element ($\boldsymbol{x}_a$) is selected at each step $j$, along with a latent variable ($\boldsymbol{z}_j$) sampled from a Normal distribution via learned mean and standard deviation, on which the final, per-element predictions are conditioned for the current $j$-th cluster:

$$p(\boldsymbol{z}_j \mid \mathbb{X}_j) = \mathcal{N}(\boldsymbol{z}_j \mid \boldsymbol{x}_a, \boldsymbol{U}_j, \boldsymbol{G}_j), \tag{3}$$

$$p_{\theta,i}(\hat{c}_i = 1 \mid \mathbb{X}_j) = \text{sigmoid}(\rho(\boldsymbol{x}_i, \boldsymbol{x}_a, \boldsymbol{z}_j, \boldsymbol{U}_j, \boldsymbol{G}_j)). \tag{4}$$

Thirdly, **pointer attention** can be used to select a single element from a set of any cardinality $n$, common in set-to-sequence NNs. At each step $m \in \{1, \dots, k\}$ it outputs an attention vector ($\boldsymbol{a}_m$) over all obtained clusters $\mathbb{C}$. As the clusters are selected sequentially, this represents their predicted order, with highest attention value pointing to the index of the $m$-th cluster in that order:

$$\boldsymbol{a}_m = \text{softmax}(\boldsymbol{v}^\top \tanh(\boldsymbol{W}_1 \mathbb{C} + \boldsymbol{W}_2 \boldsymbol{h}_m^d)), \tag{5}$$

where $\boldsymbol{v}$, $\boldsymbol{W}_1$ and $\boldsymbol{W}_2$ are model parameters, $\tanh$ is the hyperbolic tangent nonlinearity, and $\boldsymbol{h}_m^d$ is customarily the hidden state of the pointer network at current selection step $m$. The first hidden state $\boldsymbol{h}_0^d$ can be initialized from the permutation invariant set representation $\boldsymbol{s}$. In our context, this in principle enables us to sequentially select predicted clusters according to their learned target order.

## 4 THE NEURAL ORDERED CLUSTERS MODEL

Existing methods do not, to the best of our knowledge, directly address the catalog problem of joint clustering and cluster ordering. To this end, we investigate a set of novel and adapted methods to apply to this problem. This section introduces the proposed Neural Ordered Clusters (NOC) model. NOC consists of three modular parts, each with a corresponding loss factor. These components take the form of partitional neural clustering, per-cluster cardinality prediction, and cluster ordering via pointer attention. The learned representations of elements and the set in its entirety are transformed by each of these modules and continuously adjusted during training in a fully differentiable way. For an overview of the NOC architecture, we refer the reader to Figure 2.

The **first step** is to obtain a partitional clustering (NOC$_1$). We propose to achieve this through an adjusted neural clustering module, building on the process described in equations 3 and 4. First, we utilize the *Set Interdependence Transformer*, or SIT (Jurewicz & Derczynski, 2022), to obtain both the representations of the individual elements ($\boldsymbol{E}_\pi$) and the permutation-invariant representation of the entire set ($\boldsymbol{s}$). SIT consists of a stack of MAB layers described in equations 1 and 2, except that the second layer's input takes the form of an augmented matrix, in which the vector representation of the set is concatenated to $\boldsymbol{E}_\pi$ as if it was an additional set element $\boldsymbol{e}_i$. This is intended to enable learning of higher-order interactions in fewer layers. At each cluster prediction step $j$ the representations of unassigned elements ($\boldsymbol{U}_j$) and each previously completed cluster ($\boldsymbol{g}_{1:j}$) are adjusted through a stack of SIT transformations and used to make cluster assignments $\hat{c}_i$ per unassigned element $i$:

$$\text{NOC}_1(e_i, \mathbb{X}_j) = p_{\theta,i}(\hat{c}_i = 1 \mid \mathbb{X}_j) = \sigma(\phi_1(\boldsymbol{e}_i, \boldsymbol{e}_a^j, \boldsymbol{z}_j, \text{SIT}(\boldsymbol{U}_j), \text{SIT}(\boldsymbol{g}_{1:j}))). \tag{6}$$

The **second step** (NOC$_2$) is to adjust the cluster assignments via the predicted cardinality $t_j$ of the $j$-th cluster. At each step a function, parameterized by a fully-connected neural network $\phi_2$, is used to predict the cardinality of the current cluster as a regression task. The obtained cardinality, conditioned on the available elements and previously predicted clusters, is used as a threshold for the maximum number of elements to assign to current cluster. If the number of elements assigned by NOC$_1$ exceeds this threshold, the elements with lower values of $\hat{c}_i$ are excluded from cluster $\mathbb{C}_j$.

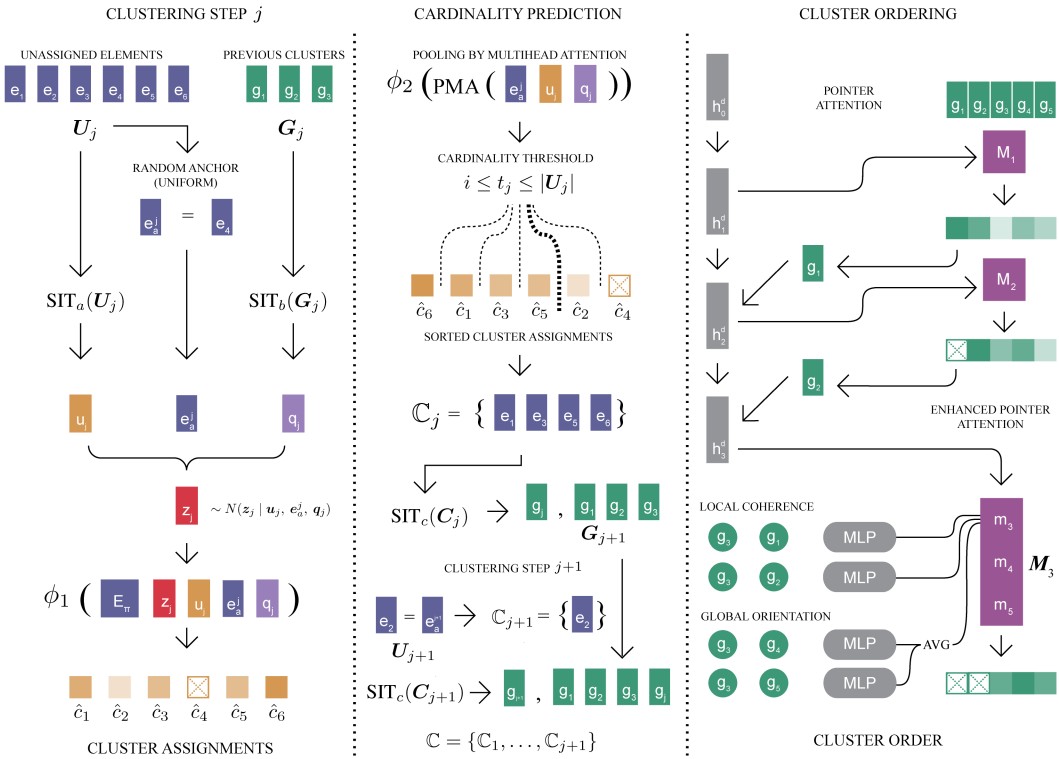

Figure 2: NOC. Starting at the top of the leftmost panel, at clustering step $j$ the representations of unassigned elements ($U_j$), previously created clusters ($G_j$) and a random anchor element ($e_a^j$) are used to make initial cluster assignments ($\hat{c}_{1-6}$). In the middle panel the current cardinality ($t_j$) is predicted and used to adjust the $j^{\text{th}}$ cluster, which is then transformed via $\text{SIT}_c(C_j)$ into its embedded representation $g_j$, which becomes part of the $G_{j+1}$ matrix and is used during the remaining clustering steps. In the rightmost panel, after $k$ iterations of the $\text{NOC}_1$ and $\text{NOC}_2$ steps, the predicted clusters are ordered via $\text{NOC}_3$'s Enhanced Pointer attention (A.2).

$$t_j = \text{NOC}_2(\mathbb{C}_j, \mathbb{X}_j) = \phi_2(\text{PMA}(e_a^j, \text{SIT}(U_j), \text{SIT}(g_{1:j}))), \tag{7}$$

$$\mathbb{C}_j = \begin{cases} C^j, & \text{if } |\mathbb{C}_j| \leq t_j \\ C_{1:t_j}^j, & \text{otherwise} \end{cases}. \tag{8}$$

Steps one and two are repeated until we have obtained $k$ partitional clusters ($\mathbb{C}_1, \ldots \mathbb{C}_k$) with individual cardinalities. Set-to-sequence methods expect fixed-length vector representations, therefore we use SIT and PMA to obtain them ($C_\pi = [c_1, \ldots, c_k]$ where $c_j = \text{PMA}(\text{SIT}(\mathbb{C}_j))$). In the **third and final stage** of $\text{NOC}_3$ an Enhanced Pointer Network (Yin et al., 2020) is used to output an attention vector $a_m$ at each step $m \in \{1, \ldots, k\}$. The highest attention value points to the cluster to be placed at $m$-th position in the output sequence of ordered clusters:

$$a_m = \text{softmax}(v^\top \tanh(W_1 M_m + W_2 h_m^d)) \, ; \, h_m^d = \text{LSTM}(h_{m-1}^d, c_{m-1}). \tag{9}$$

This largely resembles the process outlined in Equation 5, with the exception of matrix $M_m$, specific to the Enhanced Pointer Network, explained in more detail in appendix A.2. Together, these three elements of NOC allow for the prediction of an adaptive number $k$ of partitional clusters with varying, learned cardinalities. This learning is achieved through a weighted sum of the loss factors from each of the three stages of NOC, with teacher-forcing (Williams & Zipser, 1989). The full algorithm is provided in Appendix A.1.

Table 1: Clustering and permutation results on all three datasets

| Method | 2D Gaussians | | Procedural Catalogs | | PROCAT | |
|---|---|---|---|---|---|---|
| | V-Measure | Kendall's $\tau$ | V-Measure | Kendall's $\tau$ | V-Measure | Kendall's $\tau$ |
| NCP + S2S | 91.52 ± 3.30 | 75.31 ± 4.5 | 63.12 ± 4.12 | 74.82 ± 5.1 | 25.42 ± 5.14 | 21.94 ± 4.3 |
| CCP + S2S | 93.94 ± 2.13 | 83.88 ± 4.2 | 79.41 ± 3.76 | 81.10 ± 3.9 | 37.41 ± 3.10 | 25.24 ± 4.0 |
| ACP + S2S | 96.63 ± 1.82 | 90.13 ± 3.7 | 87.66 ± 3.91 | 85.73 ± 3.2 | 41.38 ± 3.88 | 31.73 ± 3.1 |
| S2S-B | 89.37 ± 4.21 | 95.89 ± 2.3 | 78.39 ± 1.64 | 92.13 ± 2.0 | 39.01 ± 3.35 | 44.39 ± 3.7 |
| S2S-C | 92.45 ± 2.01 | 93.41 ± 2.1 | 75.83 ± 4.91 | 91.55 ± 3.3 | 36.71 ± 4.26 | 40.22 ± 4.2 |
| NOC | **97.81 ± 0.92** | **98.40 ± 0.5** | **96.13 ± 1.28** | **95.84 ± 0.9** | **52.84 ± 3.15** | **56.67 ± 2.8** |

## 5 EXPERIMENTS

The Catalog Problem presents an interesting type signature, where while the input — as in S2S — is an unordered set, the output is a more complex structure that is the result of clustering and ordering. In this section we examine multiple approaches to the Catalog Problem, including baseline methods adapted to this output structure as well the NOC model, evaluating over both synthetic and real-world datasets. All datasets, hyperparameters and code are freely available and described in detail in Appendix A.4. The provided code includes all data pre-processing and generation steps.

The models' exact layer dimensions are given in Appendix A.4, with the number of learnable parameters of each model varying by less than 5% per task. The AdamW (Loshchilov & Hutter, 2017) optimizer was used with weight decay coefficient 1e-3, learning rate ($\alpha$) 1e-4, dropout rate of 0.05 and batch size 64, for 50–100 epochs. Experiments were performed on cloud-based GPU instances, with NVIDIA Quadro P6000 graphics cards (24 GB) and 8 CPU cores. To represent natural language entities in Section 5.4 we use the concatenated and averaged output of the last 4 layers of the cased, large version of BERT (Devlin et al., 2019), frozen during training to isolate the effect of compared clustering and permutation methods on the final performance.

The best performance is reported in **bold** and second best is underlined. Reported results are averaged over three full training runs, standard deviation is reported after the ± sign. We use V-Measure (Rosenberg & Hirschberg, 2007) and Kendall's Rank Correlation Coefficient ($\tau$) as the primary clustering and permutation metrics respectively, scaled by a factor of a hundred for readability, following convention (Wang & Wan, 2019; Pandey & Chowdary, 2020).

### 5.1 BASELINES

We present two groups of baselines for addressing the Catalog Problem. **i) Neural clustering methods with an added set-to-sequence module:** the module takes the predicted clusters and outputs their order via attention-based pointing. These methods include the pointwise Neural Clustering Process (NCP), the Clusterwise Clustering Process (CCP) and the Attentive Clustering Process (ACP) developed by Pakman et al. (2020) and Wang et al. (2021). **ii) Proposed variants of the set-to-sequence architecture:** these S2S variants enhance the pointer mechanism with the notion of predicting ordered *clusters*, as opposed to ordered *elements*. The first variant, called S2S-B (for "break"), adds a secondary decision of whether or not to start a new cluster in parallel to the selection of the set element to be placed next in the predicted sequence. The second variant, called S2S-C (for "clusterwise"), uses a threshold mechanism to select multiple elements forming a single cluster at each step. For details, see Appendix A.3.

### 5.2 ORDERED MIXTURES OF GAUSSIANS

This dataset consists of 2D coordinates for a number of points, generated from a mixture of a finite number of Gaussian distributions. The points should be clustered and the clusters ordered by distance from origin. Following convention from probabilistic models for clustering (McLachlan & Basford, 1988), we introduce a random variable $c_i$ signifying the cluster to which each data point $x_i$ is assigned. The generation process creates a random number of clusters $k$, each with their

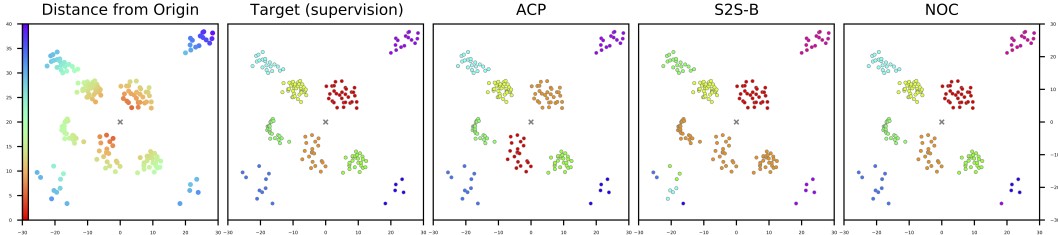

Figure 3: Example of predicted ordered clusters. Target (supervision) shows clusters and their order through colour, red being closest to the origin point (marked with a gray ×), dark blue and violet being furthest. Heat map (leftmost) indicates distance for individual points. The ACP prediction exhibits good clustering, but errs in the ordering (mistaken red and orange clusters). S2S-B exhibits good ordering, but incorrect clustering in the bottom-left quarter. NOC (ours) is closest to the target.

own parameter vector $\boldsymbol{\mu}_j$ controlling the distribution of the $j$-th cluster. For comparison with prior work (Pakman et al., 2020), we use a Chinese Restaurant Process with a single modification — the addition of a target order of the clusters, based on the Euclidean distance of their *centroids* from the origin point. An example of the joint prediction of per-element cluster assignments and predicted cluster order can be seen in Figure 3. The predicted order is denoted through colour gradient, with a bright red to deep blue and violet scale. In the figure, three separate predictions are displayed, one from the ACP model, one from a modification of set-to-sequence methods in the form of S2S-B and finally one from the proposed NOC model.

As shown in the rightmost column of Table 1, NOC outperforms other methods on both the clustering task, according to V-Measure, and the cluster ordering task, measured with Kendall's $\tau$. Specifically it improves by +1.18 points over the second-best clustering method (ACP) and +2.51 over the second-best set-to-sequence method (S2S-B). Its performance appears relatively consistent, showing a smaller standard deviation over three full training runs.

## 5.3 PROCEDURALLY GENERATED CATALOGS

The second experiment uses synthetic catalogs. These catalogs consist of varying-length sequences of clusters of elements, with repetition. Elements are colour-coded. These catalogs form the supervised training targets $\boldsymbol{y}_i$, with the unordered set of available atomic elements forming the inputs $\mathbb{X}_i$. The correct composition of individual sections and the structure of the overall catalog, in the form of the order of its sections, depends on $n$-th order interactions between the input elements.

For procedural generation, these interactions are formalized as compositional (intra-cluster) and structural (inter-cluster) rules. A simplified example of a compositional rule would be: *"if the input set contains only red, blue and yellow elements, a section containing red and yellow elements in 1:1 ratio is a valid section"*. An example of a structural rule could specify that (given the same input) the catalog has to begin with an all-red section or end on an all-blue one (top row of Figure 4). Compositional rules also include upper cardinality constraints for valid sections. We use the tool provided by Jurewicz & Derczynski (2022) to generate catalogs.

As shown in Table 2, neural clustering methods appear to be better at composing valid catalog sections but struggle with ordering sections into valid catalogs. This is indicated through two metrics – a *compositional score*, which is the percentage of predicted sections that were valid in accordance with the applicable $n$-th order ruleset, and a *structural score*, which is the percentage of valid predicted catalog structures (i.e. section ordering). By contrast, the adapted S2S models outperform neural clustering methods at correctly ordering sections, as measured via the structural score. NOC outperforms both methods on each of the two scores. This improvement is also reflected on the same test set in the more general but related V-Measure and Kendall's $\tau$, shown in Table 1, where NOC surpasses the next-best models by +8.47 and +3.71 percentage points.

Among the sections predicted by neural clustering methods (NCP, CCP, ACP), the predominant error (present in 74% of invalid sections) stemmed from incorrect cardinality, even though the models correctly predict the composition (15%) and ratio (11%) of elements to include. This error occurs despite the presence of mechanisms that could, in principle, allow for the learning of max-cardinality

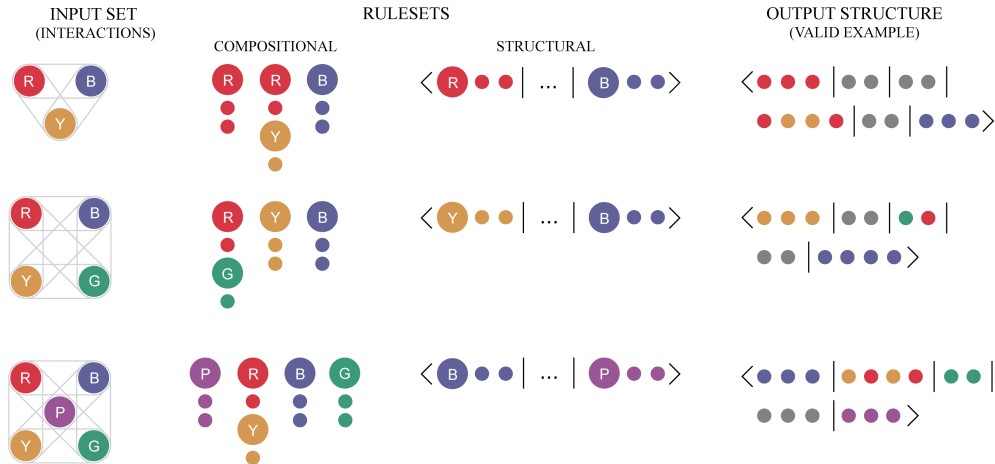

Figure 4: Procedurally generated catalogs. Relations between elements of the input set define the compositional and structural rules, which inform the generation of these synthetic datasets. A successful model should learn these rules from supervised exposure to the resulting synthetic datasets, and then be able to order new sets of elements according to the learned rules. One valid example is given for each input set (wrapped over 2 lines). See the second paragraph of Section 5.3 for a written description of the compositional and structural ruleset portrayed in the top row.

constraints: NCP constructs clusters element-by-element, further transforming the candidate cluster at each element's addition; CCP and ACP obtain a representation of the current candidate cluster before assigning candidate elements.

NOC overcomes this limitation through the addition of a cluster-level cardinality prediction mechanism and corresponding loss. It outperforms the second best method on the section composition task by +11.96, +13.01 and +15.65 percentage points with regards to the 3rd, 4th and 5th order relational ruleset respectively. It also performs better with regards to the structural score, offering a smaller but consistent improvement over the S2S-C and S2S-B methods by +4.54, +3.59 and +3.61 points, with respect to increasing $n$-th order rulesets.

Table 2: Results over procedurally generated catalogs, by $n$-th order relational ruleset

| | Compositional score | | | Structural score | | |
|---|---|---|---|---|---|---|
| Method | $n = 3$ | $n = 4$ | $n = 5$ | $n = 3$ | $n = 4$ | $n = 5$ |
| NCP + S2S | 64.13 ± 3.9 | 55.81 ± 4.6 | 51.82 ± 5.2 | 56.49 ± 4.0 | 51.87 ± 5.1 | 49.70 ± 6.8 |
| CCP + S2S | 75.40 ± 3.2 | 71.49 ± 4.3 | 65.11 ± 4.5 | 70.21 ± 3.5 | 68.39 ± 4.7 | 66.55 ± 5.4 |
| ACP + S2S | 87.05 ± 1.7 | 81.33 ± 1.9 | 76.83 ± 2.2 | 81.09 ± 2.2 | 76.34 ± 3.4 | 73.86 ± 3.8 |
| S2S-B | 84.99 ± 0.5 | 82.90 ± 0.7 | 74.82 ± 0.6 | 92.33 ± 1.5 | 89.83 ± 2.1 | 87.31 ± 2.0 |
| S2S-C | 82.03 ± 1.8 | 78.74 ± 2.1 | 72.13 ± 2.4 | 92.49 ± 1.6 | 87.41 ± 2.2 | 85.05 ± 2.3 |
| NOC | **99.01 ± 0.3** | **95.91 ± 0.4** | **92.48 ± 0.4** | **97.03 ± 0.9** | **93.42 ± 1.0** | **90.92 ± 1.2** |

## 5.4 PROCAT

The final experiment was performed on the PROCAT dataset (Jurewicz & Derczynski, 2021), using its provided training and testing split. All models were trained on approximately 9K product catalogs and tested on a separate set of 2K catalogs. Unlike the benchmarks provided with the PROCAT dataset, this formulation of the task mirrors the Catalog Problem exactly, with no information about the target number of sections (clusters) being available to the models. Individual elements were transformed into vector representations via a pre-trained, frozen language model as described at the beginning of Section 5, removing its effect on the variation in performance on the downstream task. Figure 5 displays a sample catalog predicted by NOC from a PROCAT input set of product offers.

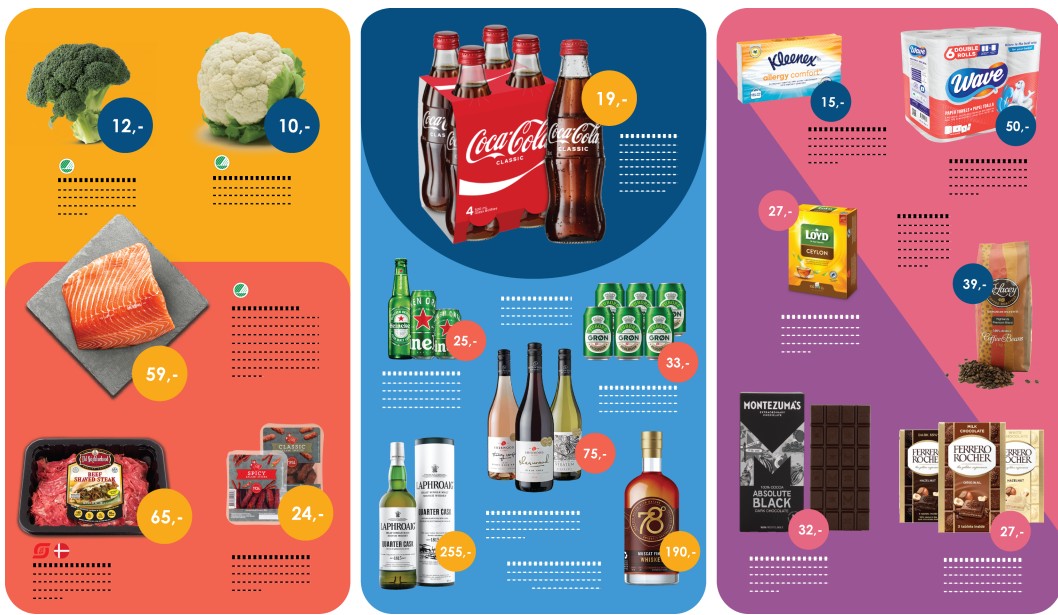

Figure 5: PROCAT. An example of three sequential sections predicted by the NOC as part of a larger catalog, from an input set of products from the PROCAT dataset. The prediction groups elements into complementary sections (the three pages shown above) and orders them into a rendered catalog.

As the two rightmost columns of Table 1 show, the PROCAT structure prediction task is more difficult than the previous tasks. The best results in terms of both the clustering quality (via V-Measure) and section order (measured indirectly via Kendall's $\tau$ with regards to element order) are approximately 40% below the corresponding scores on the procedural task in its default configuration. One possible explanation stems from the existence of a higher number of reasonable substitutions for each element in any given section from the entire input set of initially available products. While also present in the procedural catalogs, this challenge becomes harder to overcome as the cardinality of the input increases from tens in the procedural case to hundreds in PROCAT.

NOC outperforms both neural clustering methods and the adjusted set-to-sequence models. While the overall pattern of neural clustering methods outperforming S2S-B and S2S-C in V-Measure is upheld, it is less pronounced (+2.37 points between ACP and S2S-B on PROCAT compared to +9.27 and +4.18 on the procedural and 2D Gaussian task respectively). The adjusted set-to-sequence models continue to outperform NCP, CCP and ACP on the ordering aspect of the task, with a margin of +12.66 points. NOC yields the best performance in terms of both partitional clustering and ordering, exceeding the relevant second-best methods by +11.46% and +12.28% respectively.

## 6 CONCLUSION

The posited Catalog Problem consists of learning to group elements and to order the groups. It poses a more difficult challenge than its individual components. Our work defined benchmark tasks representing this problem and presented approaches for them, including both adjusted baselines and a candidate approach, Neural Ordered Clusters (NOC). Existing neural clustering methods appear ineffective at learning cluster-level cardinality constraints. Our method offers an improvement in this area through its cardinality-prediction module. NOC outperforms adjusted S2S methods in terms of both clustering quality and accuracy of the predicted cluster order, indicating that structuring models to address adaptive ordered clustering leads to improved performance over standard S2S prediction.

Nevertheless, the complexity and fluidity of intra- and inter-cluster relations result in the Catalog Problem remaining significantly more challenging than S2S processing. We considered a predictive solution to the catalog problem, where we trained the model to yield a single "ground truth" human-generated catalog given a set of products. Future work could consider a fully generative formulation of the problem that respects an unlimited number of valid solutions for both clustering and ordering.

ETHICS STATEMENT

Given the e-commerce context of the third presented dataset, we must highlight the wider problem of *endless scroll* user interfaces in product presentation apps and social media (Lupinacci Amaral, 2020). Although the PROCAT dataset is tailored to the prediction of cluster sequences of finite lengths, we cannot rule out the possibility of extending the proposed adaptive clustering and cluster ordering models to non-finite sets. It is also in principle possible to retrain the proposed models with additional inputs such as embedded personal preferences, making the predicted catalogs tailored to specific individuals, which has previously been linked to mental health issues in relation to smartphone addiction (Noë et al., 2019).

As with many machine learning systems, the results are not perfect, and sub-optimal predictions from NOC could silently disadvantage an end-user; for example a business may produce catalogs that don't make it easier for the reader to discover relevant, cost-saving offers, or an individual may receive an inaccurate medical analysis (in the case of the hypothesized medical triage use case).

Applying this tool may impact the employment of people performing creative catalog-related tasks, and further, might not even do the task as well as them. Product catalog design is considered something of an art among its practitioners, and there may be deep interactions not clearly evinced in training data that are lost by transiting the ownership of the catalog construction task from human subject matter experts to a machine learning model. Attempting to completely replace a human at this task may lead to both unsatisfactory and marginalizing results Birhane (2021).

We do not see any direct way for the presented methods to exacerbate bias against people of a certain gender, race, sexuality, or who have other protected characteristics. However, bias inherent to the marketing decisions made by people who have designed the catalogues contained in the PROCAT dataset, will be propagated by models trained on it. Negative biases in this particular scenario include as the *pink tax* (Stevens & Shanahan, 2017). In general, learning from socially-biased data and making predictions based on it will propagate those biases Buolamwini (2017); Raji (2020).

REPRODUCIBILITY STATEMENT

In order to ensure reproducibility all code and datasets needed for repeated experiments have been made freely available, as described in detail in Appendix A.4 as part of the provided supplementary materials. The anonymized code repository includes a comprehensive readme.md file describing the necessary steps to set up the execution environment, download, generate and preprocess the datasets and run each of the experiments discussed in Section 5. The exact hyperparameters per experiment are stated both in the Appendix (A.4.1, A.4.2, A.4.3) and in the provided configuration files in the linked code repository. Additionally, a detailed description of the NOC algorithm is provided (1) to ensure that the method can be reimplemented, if necessary.

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

# A APPENDIX

## A.1 NOC ALGORITHM

In this section of the appendix we outline the progression over all three stages of the proposed Neural Ordered Clusters method. Steps 1-23 jointly describe the processing within $NOC_1$ and $NOC_2$ (steps 12-16 specifically for the latter), as presented in Section 4. The third module, $NOC_3$ is shown in steps 24-30. We begin with an unordered set $\mathbb{X} \in \mathbb{R}^d$ (of any cardinality), and assume that it has at least two elements, which can then potentially belong to separate clusters. This set is represented as a matrix of $d$-dimensional elements, ordered according to some arbitrary permutation $\pi$ into $\boldsymbol{X}_\pi$. The initial, intermediate output comes in the form of individual clusters of elements at each $j$-th iteration, which ultimately form the set of all predicted clusters ($\mathbb{C} = \{\mathbb{C}_1, \dots, \mathbb{C}_k\}$). Each candidate cluster goes through a final cardinality prediction step, resulting in the threshold value of $t_j$, through which some elements may be excluded from their original cluster. Finally, an Enhanced Pointer Network (EPN) performs $k$ iterations, selecting a single cluster to be placed next in the final output sequence $\hat{\boldsymbol{y}}$ by the index of the highest value in the predicted attention vector $\boldsymbol{a}_m$.

---

**Algorithm 1** Neural Ordered Clusters

---

**Require:** $|\mathbb{X}| = n \geq 2$      ▷ At least two elements, otherwise single cluster
**Ensure:** $x_i \neq x_j \,\forall\, i, j \neq i \in \{1, \dots, n\}$      ▷ No repeated elements
1: $\boldsymbol{E}_\pi \leftarrow \text{SIT}(\boldsymbol{X}_\pi \sim \mathbb{X}), \; j \leftarrow 1$
2: $r \leftarrow n - 1$      ▷ Track number of unassigned elements
3: $\boldsymbol{e}_a^j \leftarrow \boldsymbol{E}_\pi$      ▷ Randomly chosen anchor for initial cluster
4: $\boldsymbol{U}_j \leftarrow \text{SIT}(\mathbb{E} \setminus \{\boldsymbol{e}_a^j\})$      ▷ Initialize unassigned representations
5: $\boldsymbol{q}_j \leftarrow \varnothing$      ▷ No previous clusters
6: **while** $r > 1$ **do**
7:      $\boldsymbol{z}_j \sim \mathcal{N}(\text{SIT}(\boldsymbol{e}_a^j, \boldsymbol{U}_j, \boldsymbol{g}_{1:j}))$
8:      **for** $i \leftarrow 1 \dots r$ **do**
9:          $\hat{c}_i \leftarrow \phi_1(\boldsymbol{e}_i, \boldsymbol{e}_a^j, \boldsymbol{z}_j, \text{SIT}(\boldsymbol{U}_j), \text{SIT}(\boldsymbol{g}_{1:j}))$      ▷ $j$-th cluster assignments per element
10:      **end for**
11:      $\boldsymbol{C}_j \leftarrow \boldsymbol{E}_{i:\hat{c}_i=1}^\pi$      ▷ Cluster $j$ from assignments (sorted)
12:      $t_j \leftarrow \phi_2(\text{PMA}(\boldsymbol{e}_a^j, \text{SIT}(\boldsymbol{U}_j), \text{SIT}(\boldsymbol{g}_{1:j})))$      ▷ Predict cluster cardinality
13:      **if** $|\mathbb{C}_j| \leq t_j$ **then**
14:          $\mathbb{C}_j \leftarrow \boldsymbol{C}_j$
15:      **else**
16:          $\mathbb{C}_j \leftarrow \boldsymbol{C}_{1:t_j}$      ▷ Adjust $j$-th cluster's cardinality
17:      **end if**
18:      $j \leftarrow j + 1$
19:      $\boldsymbol{e}_a^j \leftarrow \boldsymbol{E}_\pi$      ▷ Randomly chosen anchor for next cluster
20:      $\boldsymbol{U}_j \leftarrow \text{SIT}(\mathbb{E} \setminus \mathbb{C}_j)$      ▷ Update unassigned representations
21:      $\boldsymbol{q}_j \leftarrow \mathbb{C} = \{\mathbb{C}_1, \dots, \mathbb{C}_j\}$      ▷ Update preceding clusters' representations
22:      $r \leftarrow r - |\mathbb{C}_j| - 1$      ▷ Adjust number of unassigned elements
23: **end while**
24: $\boldsymbol{h}_1^d \leftarrow \text{SIT}(\boldsymbol{C}_\pi \sim \mathbb{C})$      ▷ First hidden state from all clusters
25: $\hat{\boldsymbol{y}} = (\varnothing_1, \dots, \varnothing_j)$      ▷ Final prediction placeholder
26: **for** $m \leftarrow 1 \dots k$ **do**
27:      $\boldsymbol{a}_m, \boldsymbol{h}_m^d \leftarrow \text{EPN}(\boldsymbol{C}_\pi, \boldsymbol{h}_m^d)$      ▷ Enhanced pointer attention over $k$ predicted clusters
28:      $l \leftarrow \arg\max(\boldsymbol{a}_m)$
29:      $\hat{y}_m \leftarrow \mathbb{C}_l$      ▷ Next cluster by highest attention index
30: **end for**

---

## A.2 ENHANCED POINTER NETWORK

In all reported experiments we use the same set-to-sequence module, the Enhanced Pointer Network Yin et al. (2020), which is a pointer-attention based method inspired by the popular Pointer Network Vinyals et al. (2015). It offers a performance improvement by leveraging two additional mecha-

nisms for pairwise ordering predictions towards improved global and local coherence of the output sequence. Formally, the conditional probability of a predicted order $\hat{y}$ is calculated as:

$$p_\theta(\hat{y} \mid \mathbb{C}) = \prod_{j=1}^{K} p_\theta(\hat{y_j} \mid \hat{y}_{<j}, C_\pi, s_c) \; ; \; s_c = \text{PMA}(\text{SIT}(C_\pi)) \tag{10}$$

$$p_\theta(\hat{y_j} \mid \hat{y}_{<j}, \mathbb{C}) = \text{softmax}(v^\top \tanh(W_1 h_j^d + W_2 M_j)) \tag{11}$$

$$h_j^d = \text{LSTM}(h_{j-1}^d, c_{j-1}) \; , \; h_0^d = s_c \tag{12}$$

where $v$, $W_1$ and $W_2$ are model parameters, $K$ is the total number of clusters, tanh is the hyperbolic tangent nonlinearity, $c_{j-1}$ is the fixed-length embedding of the cluster selected at the preceding step $j-1$ and $h_j^d$ is the hidden state of the permutation module at current step $i$. The first hidden state $h_0^d$ is initialized from the permutation invariant set representation of all previously predicted clusters $s_c$, obtained via SIT and PMA. The $M_j$ matrix provides additional context consisting of 2 kinds of information. The first is global orientation relating all remaining unordered clusters to one another. The second is local coherence between previously selected clusters and remaining candidates. This contextual information is obtained via *history* and *future* sub-modules from the original matrix of all cluster representations ($C_\pi \approx \mathbb{C}$). These two sub-modules output pairwise ordering predictions in relation to each candidate cluster, which are then concatenated to form $M_i$. For exact implementation details, we refer the reader to Yin et al. (2020).

## A.3 SET-TO-SEQUENCE BASELINES

In this subsection, a more detailed description of the proposed S2S variants is given. The **S2S-B variant** utilizes pointer attention to select individual remaining set elements at each step, following the convention of Pointer Networks Vinyals et al. (2015) and their enhancements (Yin et al., 2020). What distinguishes S2S-B from these models is an added prediction target which requires making $n-1$ binary decisions, where $n$ is the cardinality of the input set. At each step of the predicted permutation sequence, S2S-B indicates whether the currently selected element should be the last one of the current, open cluster. If so, this would indicate a *"break"* in the sequence, reminiscent of a page break in a product catalog. Once the last available element is reached, any remaining opened clusters are closed by default, hence $n-1$. All previously pointed-to elements since the last break are considered members of the current open cluster.

The S2S-B model is thus capable of predicting a clustering where each element is assigned its own cluster and one where all elements belong to a single cluster. It is guaranteed to assign a cluster to every single element and can handle varying cardinality input sets, like all pointer networks. The first difficulty faced due to this particular modification stems from highly skewed class distribution. Namely, we never complete (or break) a cluster after each element. This is mitigated via a class-weighted binary cross-entropy loss function:

$$\mathcal{L}_{\text{BCE-w}}(\theta) = -\frac{1}{m}\sum_{i=1}^{m}(w_b \times y_m \times \log(\hat{y}_m) + (1 - y_m) \times \log(1 - \hat{y}_m) \tag{13}$$

Where $m$ is the number of training examples, $w_b$ is the adjusted weight for the positive class, and $y_i$ and $\hat{y}_i$ are the target and prediction respectably. This loss factor is then scaled and added to the negative loss likelihood loss used to train the pointer selection mechanism. The main disadvantage of this model is that it predicts meaningless in-cluster order, making the loss signal noisy. The order of elements within each cluster is meaningless within the confines of the presented Catalog Problem.

To mitigate this disadvantage, a second variant was developed. Referred to as **S2S-C** (for "cluster-wise"), this model predicts the entire next cluster of elements at each step, instead of pointing to a single next element in the output sequence. Instead of performing $n$ transformation steps in a loop over the entire input set, it outputs an attention vector over all available elements until there are none

left. Thus it is also bound between assigning all elements to a single cluster or every element to its own cluster, much like S2S-B, guaranteeing cluster assignment for each element of the input set.

In order to predict clusters of adaptive, input-dependent cardinality, the formula for obtaining the pointer-attention vector over available elements had to be adjusted. The softmax operator was replaced with the sigmoid function ($\sigma$) and a threshold ($t_a$) of 0.5 was adopted. At each step $j \in \{1, 2 \ldots, n\}$ every element with a corresponding attention value ($a_i^j$) above the threshold is thus assigned to the next cluster:

$$\boldsymbol{a}_i = \sigma(\boldsymbol{v}^\top \tanh(\boldsymbol{W}_2 \boldsymbol{E}_\pi + \boldsymbol{W}_1 \boldsymbol{h}_i^d)) \tag{14}$$

$$\hat{y}_i^j = \begin{cases} 0, & \text{if } a_i^j < t_a \\ 1, & \text{otherwise} \end{cases} \tag{15}$$

During training, the S2S-C model was teacher-forced (Williams & Zipser, 1989) to prevent the cascading impact of incorrect initial cluster assignment on subsequent computation steps, which is a known challenge in certain areas of machine learning, such as the multi-armed bandit problem (Gan et al., 2020). This is not a departure from the other tested models (with the exception of S2S-B), as all neural clustering baselines are also teacher-forced during training, as per author implementations of the papers that originally introduced them.

## A.4 Code, Datasets and Parameters

The code required for all three of the main experiments can be found in a fully anonymized repository under the following link:

```
https://github.com/anonymous-paper-submissions/
neural-ordered-clusters
```

Follow the instructions provided in the readme.md document to set up the necessary environment locally, via the requirements.txt file listing all necessary packages and their versions.

In the following sections we describe each dataset in more detail, including how to download or generate it. All datasets are freely available under publicly accessible links. Additionally, each section contains the specific hyperparameters used for repeated experiments as well as the exact number of layers and parameters per tested NOC model.

### A.4.1 Ordered Mixtures of 2D Gaussians

**Data**. The dataset for predicting ordered clusters of 2D Gaussians (based on their distance from the origin point) is synthetically generated when running the experiment via the linked run_gauss2D.py file. The full, default configuration is given in the parser arguments (nothing should require adjustment to run the equivalent experiment). This includes a default seed, which should help ensure repeatability. In the provided experiments we generate 30K batches of 64 examples each, for a total of just under 2 million individual training examples for a full run. Each example is a set of 5 to 100 individual points characterized by their coordinates, generated through the Chinese Restaurant Process with dispersion parameter $\alpha$ set to $0.7$ for all experiments. Unlike the batch generation process used by Pakman et al. (2020), we generate batches with diverse number of clusters and cluster cardinalities in each example.

**Hyperparameters**. The training regimen includes a learning rate adjustment from 1e-4 to 5e-5 at the 15K-th batch and 1e-5 at the 20K-th batch. The AdamW (Loshchilov & Hutter, 2017) optimizer was used with a weight decay coefficient of 1e-3. Additionally, the default weights per loss factor are provided. The main clustering loss factor $\lambda_c$ is equal to 1.0, the cluster ordering loss factor is set to $\lambda_o = 4.0$ and the cardinality prediction loss factor $\lambda_k = 3\text{e-}3$. A 100 inferences samples is generated by default during validation, final metrics being calculated for the clustering prediction with the highest probability.

**Model parameters**. The NOC model with reported performance had over 12mil trainable parameters. The element and cluster encoding functions, each consisting of three stacked ISAB layers had the input and hidden dimensions of 128. The set pooling functions consisted of two stacked

ISAB layers followed by a PMA layer, also with 128 dimensions. The $NOC_1$ clustering module consistently uses a Parametric Rectified Linear Unit (PReLU) as the nonlinearity He et al. (2015).

### A.4.2 PROCEDURALLY GENERATED CATALOGS

**Data**. The dataset for predicting the cluster composition (sections of offer tokens) and structure (order of these sections) of synthetic catalogs is automatically generated when running the linked run_synthetic.py experiment script with default parser arguments. This script loads the provided configuration file synthetic_rulesets.json which specifies all compositional and structural rulesets to which the generated synthetic catalogs will adhere. In all reported experiments we refer to this default set of rulesets, but encourage researchers to treat it as an easy-to-edit, flexible configuration that can be adjusted for other exploratory experiments.

For the experiments, we generate 300K synthetic catalogs for the training set and 75K for the validation and test sets (split into 15 data-loaders). Each example consists of 35-50 offer tokens, each batch consists of 64 examples with varied number of clusters and cluster cardinalities in each batch. The NOC model is trained over 250K batch iterations, the equivalent of 50 epochs.

**Hyperparameters**. The procedurally generated catalog training regimen includes a learning rate adjustment from 1e-4 to 5e-5 at the 100K-th batch iteration and 1e-5 at the 200K-th. The AdamW (Loshchilov & Hutter, 2017) optimizer was used with a weight decay coefficient of 1e-3. Additionally, the default weights per loss factor are provided. The main clustering loss factor $\lambda_c$ is equal to 1.0, the cluster ordering loss factor is set to $\lambda_o = 15.0$ and the cardinality prediction loss factor $\lambda_k = 0.1$. A hundred inferences samples is generated by default during validation, final metrics being calculated for the clustering prediction with the highest probability.

**Model parameters**. The NOC model with reported performance had over 18mil trainable parameters. The element and cluster encoding functions, each consisting of four stacked ISAB layers had the input and hidden dimensions of 128. The set pooling functions consisted of three stacked ISAB layers followed by a PMA layer, also with 128 dimensions. The $NOC_1$ clustering module consistently uses a Parametric Rectified Linear Unit (PReLU) as the nonlinearity He et al. (2015).

### A.4.3 PROCAT

**Data**. The PROCAT dataset is freely available under the following link:

https://figshare.com/articles/dataset/PROCAT_Product_Catalogue_
Dataset_for_Implicit_Clustering_Permutation_Learning_and_
Structure_Prediction/14709507

We follow the provided train - test split of 8K - 2K catalogs and all pre-processing steps from the original paper (Jurewicz & Derczynski, 2021). The provided section break tokens are removed in the pre-processing to enable the prediction of input-dependent number of sections. Elements are by default truncated to 512 dictionary tokens for the language-specific BERT model, available in the linked hugging face repository and the suggested max-offer threshold of 200 per catalog is followed. Batches of 64 catalogs are used. The proposed NOC model is trained for 12.5K batch-iterations, the equivalent of 100 epochs.

**Hyperparameters**. The PROCAT training regimen includes a learning rate adjustment from 1e-4 to 5e-5 at the 5K-th batch iteration and 1e-5 at the 10K-th. The AdamW (Loshchilov & Hutter, 2017) optimizer was used with a weight decay coefficient of 1e-3. Additionally, the default weights per loss factor are provided. The main clustering loss factor $\lambda_c$ is equal to 1.0, the cluster ordering loss factor is set to $\lambda_o = 10.0$ and the cardinality prediction loss factor $\lambda_k = 0.5$. A hundred inferences samples is generated by default during validation, final metrics being calculated for the clustering prediction with the highest probability.

**Model parameters**. The NOC model with reported performance had 23mil trainable parameters (not including the BERT model, which was frozen during training). The element and cluster encoding functions, each consisting of 5 stacked ISAB layers had the input and hidden dimensions of 128. The set pooling functions consisted of 4 stacked ISAB layers followed by a PMA layer, also with 128 dimensions. The $NOC_1$ clustering module consistently uses a Parametric Rectified Linear Unit (PReLU) as the nonlinearity (He et al., 2015).

