# OpenReview forum: "Clustering and Ordering Variable-Sized Sets: The Catalog Problem"
_ICLR.cc/2023/Conference — Submitted to ICLR 2023_

### Official Review · Reviewer_W2N9 · 2022-10-22

**Confidence:** 3
**Correctness:** 3
**Technical Novelty And Significance:** 3
**Empirical Novelty And Significance:** 3
**Recommendation:** 5

**Clarity, Quality, Novelty And Reproducibility:**

This manuscript contains some weaknesses in presentation of the method and key points of the experiments. It could be significantly improved with some careful explanations of details that may seem obvious to the authors but are not necessarily clear to the readers. Some of the figures, especially Figure 5 take a lot of space, yet do not add much in the way of clarifying concepts in the text.


**Strength And Weaknesses:**

The proposed method builds on state-of-the art methods for reasoning over sets using neural architectures.
However, the end result has quite a lot of machinery, as illustrated in Figure 2. This makes it unclear which components may or may not be important for overall performance. Some ablation experiments are warranted. Can some elements of the model be removed or replaced with trivial stand-ins? How would this effect performance?

The notion of previous clusters are mentioned a few times. Where do these come from? Are they the training data? Is this method iterative? There is some weakness in the explanation here.

I have a hard time thinking of examples of the catalog problem. How common is it to need to order clusters? I did not follow the references in the first paragraph, which are cited as exemplars of application areas. Is it common to have supervision for the problems in these applications. This is a weakness in the presentation and extends into the demonstration data sets. A 2-dimensional mixture of Gaussians is a toy demonstration. The description of the procedurally generated catalog is vague and difficult to appreciate. What is the ground truth ordering? Is it the order with which each cluster is generated? Similarly, there appears to be no explicit statement of what constitutes a cluster or under what principle the clusters are ordered for the PROCAT experiment. I found it difficult to appreciate the goal of the ordering on PROCAT.

**Summary Of The Paper:**

The authors present a neural architecture designed to jointly cluster observations and order the clusters. The approach is demonstrated on three data sets; two synthetic and one real-world. The approach is compared to step-wise models which first cluster and then order, as well as some additional method proposed by the authors which extend the set-to-sequence architecture. Performance is measured using V-measure and Kendall's rank correlation.


**Summary Of The Review:**

The authors address a difficult problem, yet complexity of the method and allusions to details that are not explicitly stated detract from the overall presentation.

---

### Official Review · Reviewer_Nkbm · 2022-10-24

**Confidence:** 4
**Correctness:** 3
**Technical Novelty And Significance:** 4
**Empirical Novelty And Significance:** 2
**Recommendation:** 3

**Clarity, Quality, Novelty And Reproducibility:**

I considered the proposed catalog problem since I did not know any method simultaneously obtains the clusters and rank the clusters without additional supervision. However, I doubt whether a solution to such a problem is useful in the real world because there usually exists more than one meaningful order among the groups. Without additional supervised information, the order obtained is likely to be arbitrary. For example, for the synthetic data as described in Figure 3, I am not sure why distance from the origin is a good criterion for ordering. Ordering based on either the horizontal or vertical axes also look meaningful. For the real-world dataset, as shown in Figure 5, it looks like the authors assume that the price of the items is chosen as the ground-truth quantity for the order (The authors might want to clarify what is the ground-truth ordering for this dataset ). However, it is also meaningful to items based on their weights, sizes, popularity, and expected revenues. Since more than one meaningful orders exist, I do not see how the proposed method can identify one that is useful in solving a real-world problem.

The clarity about the methods and the experiments might need to improve further. More details related to intra-set attention and multi-head self-attention block, as described in Equations (1) and (2), might need to be provided in the main text or supplementary. I do not understand Equation (3). A normal distribution is parameterized by the mean and variance. What does it mean when three quantities are provided? More detail about how Set Interdependence Transformer (SIT) is implemented should be provided. I do not understand the overall objective function of the proposed NOC method. Is the objective minimizing the reconstruction loss? The definitions for V-measure and Kendall's tau need to be provided. I would like to see the V-measure that quantifies the clustering quality when no ordering assumption is imposed. How is the ground-truth order obtained for the PROCAT dataset?


**Strength And Weaknesses:**

Strengths
The authors introduced a novel problem called the catalog problem and proposed a novel method for solving this problem.
The proposed method outperforms other methods in terms of clustering and ordering qualities.

Weaknesses
The motivation for the problem is not well described. I am not sure whether the proposed problem is realistic.
The descriptions of the methods and the experiments are not clear enough.

I have included the details in the next section.


**Summary Of The Paper:**

This paper focuses on the catalog problem that jointly separates items into groups and orders these groups. The authors proposed a novel method called Neural Ordered Clusters (NOC) to solve the problem. The proposed model is tested on synthetic and real-world datasets.


**Summary Of The Review:**

I believe the authors introduced a new problem and provided a novel method for solving the problem. However, I am not convinced that the method is helpful in solving a real-world problem. The clarity of the paper can be further improved.

---

### Official Review · Reviewer_ZYNQ · 2022-11-04

**Confidence:** 2
**Correctness:** 3
**Technical Novelty And Significance:** 3
**Empirical Novelty And Significance:** 2
**Recommendation:** 3

**Clarity, Quality, Novelty And Reproducibility:**

The clarity of the paper may need to be substantially improved.  As mentioned above, the Catalog Problem may need to be more clearly defined.  The technical part involves quite many notations but the notations are not explained clearly enough.

As the objective of the catalog problem is not clearly stated (for example, what is the ground-truth ranking of clusters), it is hard to assess the performance of the proposed methods.

The catalog problem under study appears to be new and a novel method is considered for this problem.

The technical details of the proposed method is quite concise for reproducibility.  On the other hand, the paper provides a link to the source code for reproducing the results, though I haven’t checked the code carefully.

**Strength And Weaknesses:**

Strengths:

- The paper attempts a new problem to study.
- A model based on neural network is proposed to solve the catalog problem.
- The empirical results apparently show that the proposed method performed better than other baseline methods.

Weaknesses:

- The Catalog Problem is not clearly defined in the paper.  In particular, it is not clear what information is needed for the problem.  The identified clusters are supposed to be ranked, but it is unclear if it needs any additional information for ordering the items.
- It is not clear what value the the Catalog Problem problem discussed in the paper has.  For example, products where grouped into meaning clusters in Figure 5, but it is not obvious what meaning the ordering can provide.  The proposed method may appear to be less significant if the value of underlying problem cannot be justified.
- The related work section looks a bit brief and cannot position the proposed method well among other works.


**Summary Of The Paper:**

The paper studies an apparently new problem, called Catalog Problem, which looks for grouping of items and ordering of groups.  The paper further proposes a neural network-based model called Neural Ordered Clusters for solving the problem.  The empirical results show apparently superior performance of the proposed method over several baseline methods on two synthetic data sets and one large real-world data set.


**Summary Of The Review:**

The paper proposes a new problem and a new model for solving the problem.  However, the proposed problem is not clearly defined and it cannot demonstrate the strengths of the proposed method.

---

### Official Review · Reviewer_PMJN · 2022-11-05

**Confidence:** 3
**Correctness:** 2
**Technical Novelty And Significance:** 2
**Empirical Novelty And Significance:** 2
**Recommendation:** 3

**Clarity, Quality, Novelty And Reproducibility:**

### Lack of clarity
Regrettably, I believe that the manuscript does not currently meet the standards for publication in terms of clarity. In brief:

1. The paper deals with a relatively "niche" topic, yet there is no proper problem statement section describing precisely the inputs, outputs, metrics and losses that characterise "The Catalog Problem".
2. The paper heavily relies on prior work, but Section 3 only describes the key components at a high-level and with insufficient detail to clearly delineate this paper's contributions.

To this end, I strongly encourage the authors to rewrite the paper with a focus on clarity and accounting for the fact that, arguably, the paper deals with a problem that may not be familiar to many readers. In particular:

+ The "Catalog Problem" should be mathematically defined without ambiguity.
+ All model components should be detailed in the main manuscript in an entirely self-contained manner. It should be possible for readers not familiar with prior work to understand the proposed approach end-to-end.
+ The authors should make clear exactly which model components are novel contributions and which originate from prior work.

### Lack of motivation

Arguably, the paper's main contribution is to adapt prior work to a novel task, which the authors denote "The Catalog Problem". However, the manuscript does little to convince the reader that this is a relevant problem.

To this end, I suggest the authors to:
+ Better motivate the need to not only cluster inputs, but also sort elements within each cluster.
+ Provide convince examples that this is a problem where supervised datasets can be easily collected.
+ Include more real-world examples and applications as part of the experiments.

**Details Of Ethics Concerns:**

N/A.

**Strength And Weaknesses:**

Strengths:
+ Proposed approach is reported to compare favourably to baselines in two synthetic datasets and a real-world dataset (PROCAT).

Weaknesses:
+ Severe lack of clarity throughout the paper: problem statement, method and experiments.
+ Insufficient motivation: not enough compelling evidence that "The Catalog Problem" is relevant.
+ Incremental methodological contribution: proposed approach seems a novel application of existing methods (disclaimer: this judgement may be impaired by the lack of clarity).

**Summary Of The Paper:**

This paper proposes Neural Ordered Clusters (NOC), a supervised approach for set-to-sequence prediction. The resulting model learns to cluster data *and* sort elements within each resulting cluster. NOC was evaluated on two synthetic datasets and a real-world dataset (PROCAT, Jurewickz and Derczynski 2021).

**Summary Of The Review:**

Unfortunately I do not consider the manuscript fit for publication at this stage.

The paper's main shortcoming is its severe lack of clarity which, in my opinion, necessitates a major revision to be addressed.

The paper's main contribution is arguably dealing with a rare setup: "The Catalog Problem", where items need to be *both* clustered *and* sorted within each cluster. In spite of this, I do not believe the manuscript to be sufficiently convincing with respect to the relevance and importance of this problem.

Finally, and perhaps less importantly, the manuscript's methodological contributions also appears to be limited, consisting of a novel application of existing components (c.f. Attentive Clustering Processes, Set Interdependence Transformers,  Enhanced Pointer Networks). However, this would be in my opinion only a minor problem if the authors provided sufficiently compelling evidence that this novel application is truly relevant for the field.

---

### Decision · Program_Chairs · 2023-01-20

**Decision:**

Reject

**Justification For Why Not Higher Score:**

The weaknesses of the paper that I have listed above are crucial and should be addressed for publication of this paper. Since all the reviewers agree rejecting the paper and there is no response from the authors, I reject the paper. I strongly recommend addressing all the issues raised by the reviewers for further improvement of the paper.


**Justification For Why Not Lower Score:**

N/A

**Metareview: Summary, Strengths And Weaknesses:**

This paper introduces the Catalog problem, in which one finds clusters and the order between clusters in a supervised manner, and proposes a new method for the Catalog problem, called Neural Ordered Clusters (NOC), using neural networks. The performance of the proposal is empirically examined on synthetic and real-world datasets.

### Strength

- This paper introduces a new cluster-based ordering problem.

### Weakness

- As reviewers pointed out, the clarity of this paper is not high, and significant improvement is required for publication. The Catalog problem itself is not mathematically formulated, hence it is hard to understand the paper. Also, the problem to be solved (the Catalog problem) and the proposed methodology NOC are often mixed up in the paper, which significantly deteriorates the clarity of this paper.

- Since the problem is not clearly explained, the motivation is also not clear. The authors mention that this problem is treated in a supervised manner, which means that class labels of data are available. If so, why is the clustering process required? Can one directly use the class labels and just find ordering between classes? In addition, the domain and application of the Catalog problem is also not clear.